# A Dataset for Distilling Knowledge Priors from Literature for Therapeutic Design

**Haydn Thomas Jones**[†]**, Natalie Maus, Josh Magnus Ludan,**
**Maggie Ziyu Huan, Jiaming Liang, Marcelo Der Torossian Torres, Jiatao Liang,**
**Zachary Ives, Yoseph Barash, Cesar de la Fuente-Nunez, Jacob R. Gardner**[*†]**, Mark Yatskar**[*†]

## Abstract

AI-driven discovery can greatly reduce design time and enhance new therapeutics' effectiveness. Models using simulators explore broad design spaces but risk violating implicit constraints due to a lack of experimental priors. For example, in a new analysis across diverse models on the GuacaMol benchmark using supervised classifiers, over 60% of molecules proposed had a high probability of being mutagenic. In this work, we introduce `Medex`, a dataset of priors for design problems extracted from literature describing compounds used in lab settings. It is constructed with LLM pipelines for discovering therapeutic entities in relevant paragraphs and summarizing information in concise fair-use facts. `Medex` consists of 32.3 million pairs of natural language facts, and appropriate entity representations (i.e. SMILES or RefSeq IDs). To demonstrate the potential of the data, we train LLM, CLIP, and LLaVA architectures to reason jointly about text and design targets and evaluate on tasks from the Therapeutic Data Commons (TDC). `Medex` is highly effective for creating models with strong priors: in supervised prediction problems that use our data for pretraining, our best models with 15M learnable parameters outperform larger 2B TxGemma on both regression and classification TDC tasks, and perform comparably to 9B models on average. Models built with `Medex` can be used as constraints while optimizing for novel molecules in GuacaMol, resulting in proposals that are safer and nearly as effective. We release our dataset on HuggingFace at huggingface.co/datasets/medexanon/Medex, and will provide expanded versions as the available literature grows.

## 1 Introduction

AI-driven scientific discovery within chemistry and biochemistry for therapeutic design has become one of the most exciting areas of growth for the field, with promising successes in protein folding [1, 47], antibody and *de novo* protein design [78, 73], antibiotic discovery [71], and many others. The success of these computational, data-driven approaches is fueled by the wealth and variety of publicly accessible large-scale data. Curated repositories like RCSB PDB [6], ClinVar [40], PubChem [34], UniProt [10], OAS [56], the Therapeutic Data Commons (TDC) [29], among others, have enabled easy access to data on the structure, function, and biological activity for proteins, small molecules, genetic variants, and other biological entities of interest.

Although existing datasets contain a wealth of information, they are incomplete. The majority of our knowledge of chemistry, biology, and medicine remains "locked" in natural-language text found in publications, patents, and other articles. For example, while TDC distributes small- to moderate-scale labeled datasets for specific kinds of drug safety information, the ultimate source of ground truth is found in publications, data sheets, and other human-readable resources.

---

[†] Corresponding authors: {haydnj,jacobrg,myatskar}@seas.upenn.edu, [*] equal contribution.

39th Conference on Neural Information Processing Systems (NeurIPS 2025) Track on Datasets and Benchmarks.

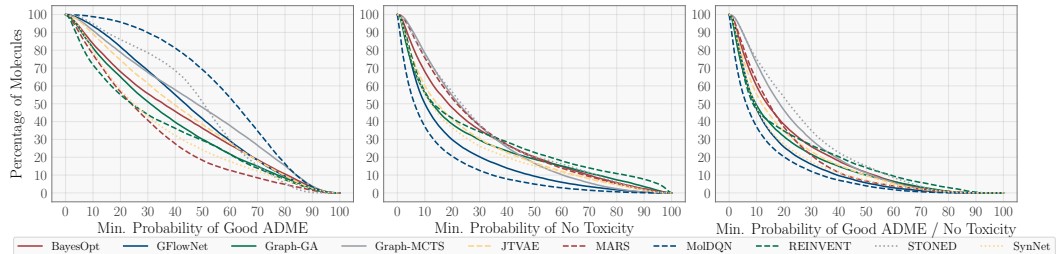

Figure 1: Frequency at which candidates are deemed *unsuitable* by our classifiers in the GuacaMol drug design benchmark [5]. If we require drug safety confidence of only $60\%$, more than $80\%$ of drugs across all methods would be removed. Methods are retrieved from [18].

Because of this relative inaccessibility of knowledge about key drug design factors like safety, stability, pharmacodynamics, and developability, many drug design benchmarks and algorithms are developed using *in silico* simulation that outright ignores these factors [53, 41, 42, 9]. To make this concrete: in Section 2, we demonstrate that a wide variety of recent works optimizing the GuacaMol benchmark suite of drug design tasks would have a large fraction of their highest scoring molecules filtered out by classifiers predicting safety characteristics considered by the TDC.

To address the lack of resources for prior knowledge relevant for therapeutic design, we present `Medex`. `Medex` is a large-scale dataset of medically relevant entities–small molecules, proteins, diseases, genes, and so on–and facts about these entities distilled from publicly accessible or licensable literature and other text sources. Our freely available dataset comprises over two million unique entities paired with information from over 200 million unique passages. For release, our dataset can be accessed as a large-scale set of succinct *facts*, along with normalized IDs and DOI sources, about the entities distilled from the literature.

`Medex` was created by leveraging recent advances in large language models (LLMs) and multimodal language modeling [49, 43, 62]. We have created and validated a mixture of supervised and zero-shot LLM components for discovering therapeutic entities in relevant paragraphs and summarizing information in concise facts. Ultimately, our pipeline offers a solution for transforming unstructured data of academic literature into tagged pairs of therapeutically relevant *entities* (small molecules, proteins, genes, and so on) linked with text found describing those entities. `Medex` can then be used in a variety of downstream multimodal models–for example using contrastive learning techniques–to build representations of entities and associated facts.

Our key contributions are as follows:

1. We release `Medex`, **a dataset of medical entities**, associated text and **32.3M extracted facts**. This data represents a significant step towards enabling machine learning models to leverage the rich biological, chemical, and medical knowledge contained in scientific literature.

2. We demonstrate the potential for our data **to greatly improve supervised and multimodal learning.** Leveraging our data, we train small multimodal models with 15M learnable parameters that outperform the larger 2B TxGemma model across TDC classification benchmark tasks, and achieve **33%** lower MAE on regression benchmark tasks. Our models perform comparably to the larger 9B parameter models on average across the TDC tasks. To further highlight the value of our knowledge extraction alone without access to additional TDC labels, we demonstrate **74% improved zero-shot performance** over baselines.

3. We demonstrate that models built with our data can be used to **constrain molecular optimization algorithms**. We optimize four Guacamol benchmark tasks using safety and toxicity constraints, and demonstrate proposals that are safer and nearly as high scoring as unconstrained solutions.

## 2 GuacaMol analysis

Many existing approaches for therapeutic candidate optimization are benchmarked primarily *in silico*: given a set of design goals expressed as a fitness function (i.e. binding affinity), the goal is to propose *de novo* high scoring molecules, often with the fewest number of tests against the fitness function [18]. Many approaches have been proposed [41, 42, 53, 9], and methods are increasingly able to produce very high scoring, high precision candidate lists. For example, many of the GuacaMol [5]

molecular design benchmark tasks can now be optimized in hundreds of evaluations [42]. Results on GuacaMol would seem to imply that, soon, given an entirely novel design problem, such methods could be used to propose a small number of candidates for scientists to test iteratively in the lab.

While results are promising and many recent papers showcase the potential of computational approaches with *in vitro* and *in vivo* data, common *in silico* benchmarks may overestimate their feasibility. The fitness functions are too narrowly defined, lacking the ability to encode diverse real-world constraints. For instance, enforcing safety constraints like low liver toxicity or prioritizing candidates with long half-lives is challenging. These practical constraints are common-sense for lab evaluations but are missing in benchmarks due to inadequate documentation and computational tools for estimating these desiderata.

To evaluate the scope of such problems, we filter the top 10% candidate molecules across all GuacaMol benchmark tasks produced by 10 methods retrieved from a meta-study [18] based on two properties: (i) low likelihood of mutagenicity and hERG channel blockade, and (ii) high absorption, distribution, metabolism, and excretion (ADME), a measure of how well a chemical is absorbed and retained in the body. Given a proposal molecule from a model, each of these properties was measured via a calibrated supervised classifier [21] trained using data from both `Medex` and TDC (see Appendix E for details). For each method, we calculated the proportion of proposals meeting specified toxicity and ADME thresholds (Figure 1). Across all methods, fewer than 10% of candidates are viable when requiring non-toxicity *or* a favorable ADME profile with 95% certainty. No proposals meet the criteria when requiring *both* at 95% certainty.

## 3 Dataset construction

Our objective is to construct a dataset of `(entity, text)` pairs that are broadly useful for conditioning machine learning models. In our context of biology, chemistry and medicine, entities consist of small molecules, proteins, genes and variants, and diseases. Figure 2 summarizes our overall approach and dataset statistics. First, documents are retrieved with help of databases of entities (Section 3.1). Entity mentions are identified in paragraphs and normalized (Section 3.2). Lastly, facts about entities are summarized from paragraphs (Section 3.3). Each processed paragraph can result in the creation of multiple facts about multiple entities. The whole process allows for attribution of facts, while normalizing entities and combining dispersed information. For example, as seen in Figure 2, we extract that Levofloxacin is "detectable in blood and brain" and relate that fact to its SMILES.

### 3.1 Sourcing relevant text

The first clear consideration is to subset the entire accessible academic literature to the papers likely to contain relevant entities. Simply processing any and all available papers is prohibitively expensive and liable to result in a very high false positive rate. To avoid this, we take an "entity-first" approach. We first collect a broad set of entities we are interested in and, for each entity (small molecule, protein, etc), we find papers that are highly likely to mention or discuss that entity.

**Entities to documents.** We use existing databases of small molecules, proteins, genes, and other entities that crucially link to papers mentioning them. For example, PubChem [34] is a repository of over 100 million compounds linking to over 40 million publications. Querying PubChem for any particular compound returns a set of PMIDs and DOIs for papers that the database claims mention that compound. For example, the PubChem page for aspirin contains links to 136,177 publications at the time of writing. Similar databases exist for other entity types, and we use UniProt [10] for proteins. In addition, one of the tagging methodologies we leverage and discuss below–most notably PubTator3 [81]–which tags small molecules, proteins, genes, gene variants, and diseases in papers, and we include the literature covered by PubTator3 in our set. In total, after joining all sources of papers, we collected a set of about 43,000,000 papers and abstracts that we consider at this stage to be *candidates* for discussing relevant entities.

**Documents to paragraphs.** After retrieving all papers that were freely or via site license accessible to us, we processed all PDFs to plain text using GROBID [20]. To separate documents into paragraphs, we use the GROBID-detected paragraph breaks, resulting in over 400 million total paragraphs.

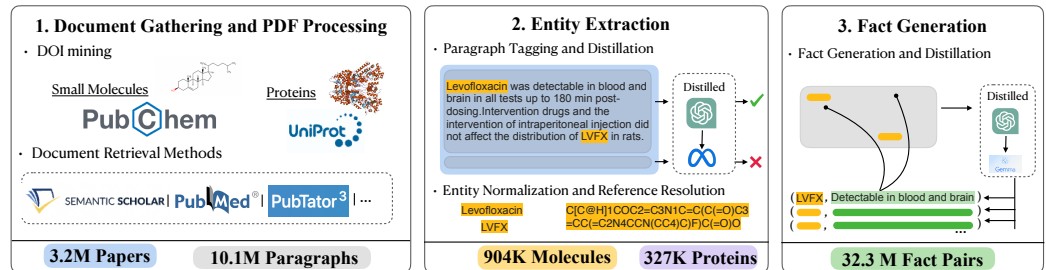

Figure 2: Our pipeline for creating pairs of therapeutic entities and natural language facts. Broadly, it is comprised of paper mining from databases (Section 3.1), entity extraction and normalization from paragraphs via distilled LLMs (Section 3.2), and fact generation via distilled LLMs (Section 3.3). Medex contains 32.3M facts about 900K molecules and 327K proteins, found in 11.2M paragraphs.

## 3.2 Tagging entities in paragraphs

Our goal is to *tag* each paragraph with its entities. We need models for this since existing databases only map entities to *papers*–not paragraphs–providing an incomplete guess of discussed entities in the paper. The decisions so far mainly affect the final *quantity* of collected data through paper selection, while accurate entity tagging of paragraphs remains one of the two key challenges impacting *quality* in dataset construction, alongside fact extraction.

In our dataset, we leverage two approaches to tagging. First, we use PubTator3[1] [81], an off-the-shelf entity tagger that tags chemicals, genes/proteins, and diseases in text. Second, we will prompt an LLM with the paragraph and ask it to identify any relevant entities described in the paragraph.

Entity tagging poses a number of challenges to consider. Here we describe three examples:

1. **Scale and expense.** At more than 400M total paragraphs needing tagging, processing this data exclusively with the highest fidelity language models is cost prohibitive at an estimated $248,000 dollars with GPT-4.1 API queries.

2. **Entity normalization.** Different papers may, when discussing the same physical chemical entity, use different names for that entity in text. For any given entity we identify in a paragraph, we need to associate that paragraph with a standardized "name" or representation of that entity.

3. **Alias resolution.** Entities in papers are often referred to by aliases. Common chemical names are abbreviated (e.g., Levofloxacin becomes LVX), and long IUPAC names are replaced with placeholders like "Compound A," resolved in tables only once in the paper. Even the acronyms or abbreviations are not always sensible in a vacuum–for example, a paper might differentiate early-onset and late-onset neonatal sepsis throughout simply by EOS and LOS, with the reference to sepsis being simply implied.

**Model distillation.** To deal with the cost of tagging, we leverage knowledge distillation. We use Llama 405B to initially tag 60,000 paragraphs with the small molecules, proteins, genes, and other entities they contain. We provide the prompts used for this tagging in Appendix A. We then use these paragraphs to distill into a Llama 3.1 8B model using LoRA [25].

To validate our knowledge distillation process, we use a curated held-out set of 3170 gold paragraphs with known tags, and evaluate the precision and recall of (1) the full Llama 405B model, (2) our fine-tuned Llama 8B model, and (3) the foundation Llama 8B model in Table 1. The precision and recall of the fine-tuned Llama 8B model approaches that of the full Llama 405B model.

**Entity normalization and alias resolution.** To normalize small molecules, we convert chemical names in paragraphs to the canonical SMILES string representation of that chemical. We use a combination of the titles of PubChem compounds, as well as the open source tool OPSIN [52] to normalize common terminology and IUPAC names. The end result after normalization is that, for each paragraph tagged with a chemical by either PubTator or our fine-tuned LLM, we will have extracted the SMILES string representation for the chemical being described if possible. If we are unable to extract a SMILES string (e.g., because our abbreviation or alias resolution fails), we discard the paragraph. To normalize

---

[1]https://www.ncbi.nlm.nih.gov/research/PubTator3/

genes/proteins, we map names to NCBI Gene IDs using Gnorm2 [82], which lets us e.g., construct amino acid sequences for gene variants. For acronym and abbreviation resolution, we use Ab3P [70].

At the end of this process, we are left with 214,000,000 tagged paragraphs, and 619,000,000 aligned (`entity, paragraph`) pairs spanning more than 2,000,000 unique entities. For more statistics, see Appendix B.

| Model | Precision | Recall |
|---|---|---|
| Llama 405B | 0.922 | 0.919 |
| Llama 8B FT | 0.905 | 0.878 |
| Llama 8B | 0.758 | 0.820 |

### 3.3 Distilling text to facts

Table 1: Precision and recall of the Llama models.

After identifying and normalizing entities in paragraphs, we extract concise factual statements about them using knowledge distillation. We use GPT-4.1 to generate facts from 60,000 paragraphs, which are used to fine-tune the more efficient Gemma 3 4B model [77]. The provided prompt specifies that a fact is a universally true, reusable property of the entity, understandable outside the paragraph's context. Examples of acceptable facts include an entity's mechanism of action, target, therapeutic or functional use, physiological role, or broader properties. The model was instructed to disregard context-lacking details (e.g., an EC50 value without assay context) and speculative statements. See Appendix A for the full prompt.

## 4 Methods and models

**Overview.** In this section, we introduce model adaptations to allow several common language modeling architectures to benefit from our fact dataset. Our approaches take inspiration from multimodal language models such as CLIP [62] and LLaVA [49], that pair images and text. We make adaptations that allow these models to pair formal representations of therapeutically relevant structures (i.e., SMILES for molecules) and text they are mentioned in. We also consider an approach that operates entirely on text. In Section 5.2, we evaluate which of these variants is most effective by considering which is best at predicting properties of therapeutically relevant tasks from TDC [29].

Our *primary* method, `MedexCLIP` (Section 4.1), learns a 128 dimensional space over structures $s \in S$ and fact text $w$ by training two 2-layer MLP adapters on top of *frozen* structure and text encoders ($E_s(\cdot), E_w(\cdot)$), and adds a 1-hidden-layer MLP head for supervised tasks. Unless noted otherwise, "our model" and the main results (Sections 5.2 to 5.5) refer to this variant. Two *alternatives* (Section 4.2), investigate other ways to use the facts: `MedexLLaVA` projects structure embeddings into an LM's token space then fine-tunes the LM, and `MedexLM` is a text-only LM instruction-tuned on fact-derived prompts. Finally, Section 4.3 introduces a **zero-shot** classifier that uses `MedexCLIP`'s space with prototype matching to isolate the information content of the facts.

**Formulation.** We define a space of possible therapeutically relevant representations (i.e. molecules represented as SMILES), $S$, where each element of $S$ can be mapped to a physical structure. Assume a fact dataset made of up of pairs of natural language statements and such representations $(w, s) \in F$, where $s \in S$ and $w$ is a sequence of words. For example, as seen in Figure 2, $F$ could contain the phrase "detectable in blood and brain" paired with the SMILES for the drug Levofloxacin.

We will also assume a set of datasets corresponding to target tasks $T$, where each $D_t \in T$ contains a set of inputs and a single output $(x_1, ..., x_k, y)$, where all $x_1, ..., x_k \in S$, and $y$ is either binary, if $t$ is classification, or real valued if $t$ is regression. Our overall goal is to construct a model that maximizes performance on tasks in $T$ by leveraging information in $F$ and samples in $D_t$.

### 4.1 Contrastively Learned Representations with Adapters (`MedexCLIP`)

The goal of `MedexCLIP` is to form a joint representation space of therapeutically relevant structures and text that co-occur with them. Such a representation will make it easy to predict features relevant to target tasks because the text expresses related properties.

**Contrastive Learning.** Given an embedding function, $E_s$ that maps any element in $S$ to $\mathbb{R}^n$, and an embedding function $E_w$ that maps any sequence of words to $\mathbb{R}^m$, we will learn to embed these features in a shared $p$-dimensional space. We learn using the standard noise-contrastive loss over $F$:

$$- \sum_{(s,w) \in F} \log \left( \frac{\exp(\phi_s(E_s(s))^T \phi_w(E_w(w))/\tau)}{\sum_{(s',w') \in F, s' \neq s} \exp(\phi_s(E_s(s))^T \phi_w(E_w(w'))/\tau)} \right)$$

Table 2: Ablation of using various multi-modal model architectures to perform supervised learning using `Medex`. While CLIP-style models perform the best, all architectures generally outperform LLMs fine-tuned with TDC data only (e.g., without `Medex`)

| Task Type | Metric | | Tasks | MedexCLIP | MedexLLaVA | MedexLM | TDC LM |
|---|---|---|---|---|---|---|---|
| Toxicity | AUROC | ↑ | 8 | **0.837** | 0.800 | 0.800 | 0.773 |
| Toxicity | Accuracy | ↑ | 2 | **0.807** | 0.798 | 0.771 | 0.750 |
| Pharmacokinetics | AUROC | ↑ | 13 | **0.830** | 0.781 | 0.824 | 0.781 |
| High-throughput screening | AUROC | ↑ | 4 | **0.737** | 0.711 | 0.718 | 0.651 |
| Clinical trial outcome | AUROC | ↑ | 3 | 0.631 | 0.636 | 0.656 | **0.683** |

Where $\phi_s : \mathbb{R}^n \to \mathbb{R}^p$ and $\phi_w : \mathbb{R}^m \to \mathbb{R}^p$ are neural networks and $\tau$ is a learned temperature parameter. The objective tries to pull co-occurring facts and structures nearby in the shared space, while pushing all pairs that do not co-occur apart. Practically, we approximate the normalization using elements in the batch.

**Adapter Heads.** The contrastive loss above allows us to learn a feature representation that aligns well with facts from literature. Finally, given task specific data from $T$, we reuse $\phi_s$ to embed inputs from all samples in the corresponding datasets. For datasets involving multiple inputs, we embed each independently and concatenate the representations, using separate heads per input signature (e.g., 1-mol, 2-mol, mol + protein, ...), each a 1-hidden-layer MLP.

## 4.2 Additional models

**Soft-prompted language models (`MedexLLaVA`).** The goal of `MedexLLaVA` is to learn to embed structures in $S$ so that they can be provided to a pretrained language model, $L$, as input. The language model can then be further adapted to reason with such structures using task-specific data.

Like previous work [49], we assume a pretrained embedding model for every element of $S$ from a CLIP model ($\phi_S$ described above). Given a language model $L$, with token embedding in $\mathbb{R}^l$, we will learn a projection function $H_S : \mathbb{R}^p \to \mathbb{R}^l$ with a neural network. In LLaVA, $H_s$ is commonly learned in an alignment phase using separate data while holding the language model frozen. `MedexLLaVA` is learned similarly, where $H_s$ is trained with a synthetically generated alignment dataset $D_a$. To generate $D_a$, we randomly sample $m \in S$ and prompt $L$ to output a string representation of m given $H_s(\phi_s(E_s(m)))$. $H_s$ is learned using a cross entropy loss, holding $L$ frozen.

**Task Adaptation.** Learning $H_S$ aligns therapeutic representations with the token embedding space of $L$. Given supervised task data in $D_t$, we map every sample to a prompt, and fine-tune $L$ with an appropriate loss for each task.

**Text-only language models (`MedexLM`).** To evaluate working entirely in text space, we create an instruction tuning dataset using `Medex`. For every fact $(w, s) \in F$, we prompt a language model to create a multiple-choice prediction question incorporating the string representation $s$. The conjecture tested here is that, while the SMILES strings of various molecules (e.g., "C[C@H]1COC2=C3N1C=C(C(=O)C3=CC(=C2N4CCN(CC4)C)F)C(=O)O" for Levofloxacin) are not human-readable, they may occur naturally during the pretraining of a large language model, and large language models may therefore be directly adaptable via instruction tuning.

## 4.3 Zero-Shot Learning

To demonstrate the information content of `Medex` in isolation, we evaluate performance *without any task–specific fine-tuning*. Following prototypical networks [69], we map each binary task in TDC to two small sets of textual descriptions and classify unseen molecules by measuring their similarity to the class prototypes.

**Prototypes.** For each TDC task, we prompt GPT-4.1 to generate ten *positive* facts ($P$) for the positive class (e.g., "*Gabapentin crosses the BBB via...*" for BBB Martins) and ten *negative* facts for the negative class (e.g., *Doxorubicin's brain uptake is limited by...*"). Each fact $w$ is embedded using the `MedexCLIP` text encoder $\phi_w(\cdot)$. Class prototypes are the averaged embeddings.

Table 3: Summary of TDC benchmark results, grouped by type. See full results tables in Appendix G.

| Task Type | Metric | | Tasks | MedexCLIP | TxGemma 2B | TDC LM |
|---|---|---|---|---|---|---|
| Toxicity | AUROC | ↑ | 8 | **0.837** | 0.822 | 0.801 |
| Toxicity | Accuracy | ↑ | 2 | **0.807** | 0.800 | 0.771 |
| Pharmacokinetics | AUROC | ↑ | 13 | **0.830** | 0.805 | 0.726 |
| High-throughput screening | AUROC | ↑ | 4 | **0.737** | 0.728 | 0.620 |
| Developability | AUPRC | ↑ | 3 | 0.659 | **0.676** | 0.616 |
| Clinical trial outcome | AUROC | ↑ | 3 | 0.631 | **0.679** | 0.661 |
| Protein interaction | AUROC | ↑ | 2 | 0.857 | **0.861** | 0.868 |
| Protein interaction | AUPRC | ↑ | 2 | 0.712 | **0.751** | 0.622 |
| Drug synergy | MAE | ↓ | 6 | 5.215 | 9.724 | **4.983** |
| Drug synergy | PCC | ↑ | 3 | **0.782** | 0.707 | 0.707 |
| Drug-target interaction | PCC | ↑ | 5 | **0.654** | 0.525 | 0.596 |
| Drug-target interaction | Spearman | ↑ | 1 | **0.813** | 0.399 | 0.548 |
| Pharmacokinetics | Spearman | ↑ | 3 | **0.472** | 0.434 | 0.359 |
| Pharmacokinetics | PCC | ↑ | 2 | 0.490 | 0.472 | **0.658** |
| Pharmacokinetics | MAE | ↓ | 3 | **3.216** | 3.612 | 3.879 |
| Reaction yields | PCC | ↑ | 1 | **0.921** | 0.661 | 0.636 |
| Reaction yields | Spearman | ↑ | 1 | 0.509 | **0.564** | 0.434 |
| Toxicity | MAE | ↓ | 1 | **0.708** | 0.71 | 0.746 |
| Developability | MAE | ↓ | 1 | **3.672** | 5.301 | 6.144 |
| Antibody affinity | MAE | ↓ | 1 | 2.338 | **1.066** | **0.968** |

**Inference.** Given test molecule $s$, its embedding is computed using `MedexCLIP`, $v = \phi_s(E_s(s))$. We compute class assignment probabilities using the inner products between molecule and prototypes:

$$\Pr(y = 1) = \sigma\left(\langle \mathbf{v}, \mathbf{z}_{\text{pos}} \rangle - \langle v, \mathbf{z}_{\text{neg}} \rangle\right) \quad \text{where} \quad \mathbf{z}_{\text{pos}} = \frac{1}{|P|} \sum_{w \in P} e_w, \quad \mathbf{z}_{\text{neg}} = \frac{1}{|N|} \sum_{w \in N} e_w,$$

where $\sigma(\cdot)$ is the sigmoid function. No parameters are updated during zero-shot inference, relying on the alignment learned during contrastive pretraining. Our prompt templates are in Appendix A.

## 5 Experimental setup and results

Our experimental setup centers on evaluating the extent to which distilling real-world priors from literature into our models can improve their predictive performance and enable safer and more effective therapeutic design. To this end, we design a set of evaluations using the diverse datasets within TDC, and a set of de novo small molecule design tasks incorporating realistic constraints. We report TDC benchmark performance (Section 5.2), zero-shot results (Section 5.3), `Medex`'s impact on predictive performance across various architectures (Section 5.4), and constrained optimization results (Section 5.5). Models are briefly outlined below, with full details available in Appendix E.

### 5.1 Models

**Structure and text encoder.** `MedexCLIP` and `MedexLLaVA` require initial embedding representations of both the entity (e.g., small molecules and proteins), and text inputs ($E_s$ and $E_w$). For small molecules, we use a T5 [63] style model trained with a masked language modeling objective. For proteins, we use ProtT5-XL [12]. Vector embeddings for molecules and proteins are produced by averaging over the sequence dimension of the encoder output. For text embeddings, we use stella_en_1.5B [89]. Both models are frozen during learning. For networks that project to the joint embedding space, $\phi_s$ and $\phi_w$, we use MLPs.

**Using `MedexCLIP` for supervised learning.** We place a 1-hidden-layer MLP on top of the joint embedding space produced by `MedexCLIP`. Architectural and training details are in Appendix E.

### 5.2 TDC Evaluation

We report quantitative results on 35 binary classification tasks and 28 regression tasks from TDC, spanning absorption, distribution, metabolism, safety, protein-protein interaction, and more. Table 3 presents a breakdown by benchmark category and full results are in Appendix G.

**Baselines.** TxGemma is the SOTA generalist that combines all TDC tasks via prompt templates and fine-tunes a Gemma model [77], outperforming many specialists. We compare against their 2B

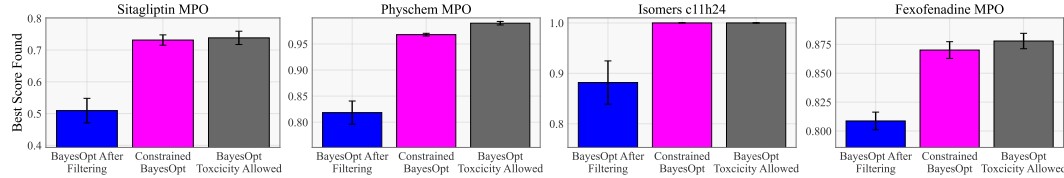

Figure 3: Bayesian optimization results on four GuacaMol tasks.

model because it has a similar total parameter count to ours. `MedexCLIP` only has 15M trainable parameters[2], so we also compare to a Qwen 500M model finetuned on TDC data (TDC LM).

**Results.** Performance on TDC tasks is summarized in Table 3. On classification tasks, `MedexCLIP` achieves an average score of $0.771$, outperforming the TxGemma-2B baseline ($0.768$) and beating out task-tailored SOTA specialist models on $10/35$ tasks. On the regression tasks, `MedexCLIP` surpasses TxGemma-2B on $23/28$ of them, while improving upon SOTA specialist methods on 12. These results show that distilling factual priors from literature can close, and often invert, the performance gap to purely supervised and specialized methods.

### 5.3 Zero-shot Evaluation

Table 4 compares `MedexCLIP` using the zero-shot learning setup described in Section 4.3 to a base 2B-parameter Gemma model that never received any supervised learning on TDC tasks. Using only self-supervised learning on `Medex`, `MedexCLIP` has a mean AUROC of $0.718$ on 9 design relevant endpoints (mutagenicity, blood brain barrier permeability, hepatoxicity, etc.), which is a $74\%$ relative improvement over Gemma-2B ($0.411$). `MedexCLIP`'s zero-shot performance is well above random, showing that the learned joint embeddings meaningfully organize molecules along textual descriptors.

| Task | Zero-shot | Gemma 2 2B |
|---|---|---|
| AMES | **0.687** | 0.487 |
| BBB Martins | **0.755** | 0.250 |
| Bioavailability | **0.596** | 0.479 |
| DILI | **0.837** | 0.320 |
| PAMPA NCATS | **0.729** | 0.465 |
| Pgp Broccatelli | **0.719** | 0.416 |
| HIA Hou | **0.852** | 0.257 |
| HIV | **0.651** | 0.491 |
| hERG | **0.634** | 0.538 |

Table 4: Zero-shot classification results on selected TDC tasks. Gemma 2 2B is used as a zero-shot baseline. All metrics are AUROC. Bold: best.

### 5.4 Architectural ablations

We evaluated supervised extensions of `MedexLLaVA` and `MedexLM` detailed in Appendix E, comparing to `MedexCLIP`. Table 2 compares these models trained on small molecule classification tasks sharing the same 1.5B backbone. While `MedexCLIP` is consistently best, every model containing distilled knowledge priors outperform TDC LM, which was finetuned exclusively on TDC. While further work is required to truly identify the architectures that best leverage `Medex`, this result underscores that the knowledge extracted in `Medex` is valuable independent of the ultimate ML model used.

### 5.5 *De novo* design with constraints

Here, we evaluate the utility of `Medex` in guiding *de novo* small molecule design towards safer candidates on the GuacaMol benchmark. Our primary analysis on existing methods in Section 2 highlighted that a significant percentage of proposed molecules are predicted to be unsafe.

In Figure 3, we run Bayesian optimization (BO) on four GuacaMol molecular design tasks [5] with a feasibility constraint: molecules must be deemed as non-toxic with $\geq 70\%$ confidence by `MedexCLIP` (i.e., non-mutagenic and not hERG-inhibiting). While this constraint can apply to other SOTA drug design methods, we leverage BO as available software directly supports black-box constraints.

Gray bars represent the best score from standard BO, allowing toxicity. Blue bars show the best score from the same BO run after removing infeasible molecules. Magenta bars represent the best score from constrained BO, which to optimizes for both feasibility and high scores. Constrained BO finds feasible molecules with scores close to those from BO allowing toxicity, even though the scores drop after filtering infeasible molecules. This result demonstrates that these GuacaMol benchmark tasks can still be well-optimized without proposing unsafe molecules. All bar plots show the mean over 20 runs and depict standard errors. See Appendix C for additional details.

---

[2]Input embeddings $E_s$ and $E_w$ are frozen

# 6   Related work

**Instruction and Fact Datasets for Molecular AI.**   Instruction-tuning datasets in chemistry and biomedicine differ in scale, language richness, and entity coverage. SMolInstruct includes 3 million examples across 14 chemistry tasks, enabling models to outperform SOTA LLMs like GPT-4 in chemistry benchmarks [87]. Mol-Instructions features 2 million biomolecular prompts on molecules, proteins, and textual biology, with many derived from ontologies [15]. DrugChat offers 143 thousand molecule-centric QA pairs for 10,834 compounds, training a GNN-LLM dialogue system [46]. MolOpt-Instructions provides 1.2 million instructions for small molecule optimization, linking a SMILES and desired property change to an improved analogue [86]. These datasets often use curated databases like PubChem [34], ChEMBL [88], or UniProt [10] for template-based examples.

**Large Language Models for Therapeutics.**   Tx-LLM used 66 TDC datasets for instruction tuning of a PaLM-2 model, achieving SOTA on 22 benchmarks and strong performance on 21 additional tasks without requiring task-specific heads [7]. Expanding this approach, TxGemma finetuned Gemma 2 [72] models (2-27B parameters) that match or surpass Tx-LLM on 64 out of 66 tasks, setting new SOTA on 45 and introducing an agentic workflow interface [77]. NatureLM integrates sequences from chemistry, biology, and materials for cross-domain generation, often equaling or outperforming specialist models in tasks like ADMET prediction [84]. MolT5 uses a text-to-text method to handle molecules and language as sequence pairs, allowing for "captioning" and prompt-driven design [11].

**Graph and Multimodal.**   Graph and multimodal encoders utilize explicit structure beyond language models. CLAMP uses contrastive learning to align PubChem BioAssay descriptions with active compounds, enabling zero-shot activity prediction, limited by brief assay texts [66] and task variety. The TxGNN knowledge-graph model, pretrained on 17k diseases and 8k drugs, enhances zero-shot repurposing by 49% and provides rationales via a multi-hop explainer [28]. MolE modifies DeBERTa [23] for molecular graphs using atom-masking and multitask pretraining, achieving SOTA on the TDC ADMET suite [55]. GIT-Mol integrates graph, image, and text inputs, boosting property-prediction accuracy by 5–10% and generation validity by 20% over unimodal baselines [50].

# 7   Conclusions

*In silico* optimization approaches for therapeutic design often produce proposals that are unsuitable for real-world use because they don't consider sufficiently broad optimization criterion. They miss prior knowledge of experimental facts that are available in scientific documents. To address this gap, we introduced `Medex`, a resource of over 32 million facts of prior knowledge relevant for therapeutic design, extracted automatically from scientific literature. We also show `Medex` is suitable as a pretraining corpus for many model architectures used today. As a result of pretraining on `Medex`, our best model, `MedexCLIP`, is extremely parameter efficient when fine-tuned on TDC data. It outperforms larger models and has set new state-of-the-art results across multiple TDC tasks. Overall, our work demonstrates that scientific documents are an untapped resource for AI-driven discovery.

**Limitations and future work.**   `Medex` is currently designed by reasoning about scientific documents independently, ignoring the larger context of the literature as a whole. This ignores the provenance and reproducibility of facts, and reputational notions of trustworthiness associated with venues or authors. Similarly, we do not carefully differentiate higher and lower certainty facts. In the future, we plan to leverage the implicit graph structure between facts across documents (e.g., corroboration by different studies) and the broader scientific literature. We will focus on enriching simple fact content with semantic links, annotations, and fused results, to provide greater context.

To our knowledge, `Medex` is among the broadest available resources for experimental prior knowledge. It can be updated as new literature becomes available, allowing for a resource that grows in both size and accuracy. It could be used to develop new architectures and pretraining approaches, and provide training data for therapeutic criteria that have no well-curated datasets.

# Acknowledgments

This research was supported by the Defense Advanced Research Projects Agency (DARPA) SciFy program (Agreement No. HR00112520300) and by the National Science Foundation (NSF) under grants DBI-2400135 and IIS-2145644. The views expressed are those of the authors and do not reflect the official policy or position of the Department of Defense, the U.S. Government, or the National Science Foundation.

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

Table 5: Aggregate statistics for the corpus that `Medex` samples from for fact extraction.

| Subset | (passage, entity) pairs | Passages | Entities | Papers |
|---|---|---|---|---|
| Small Molecules | 372,073,000 | 155,805,821 | 1,747,091 | 16,276,103 |
| Genes / Proteins | 247,323,198 | 102,869,272 | 590,747 | 11,118,371 |
| **Total** | 619,396,198 | 214,033,574 | 2,337,838 | 18,387,248 |

## A   Prompts

The prompt that was used for initial entity extraction from paragraphs using Llama 3.1 405B is shown in Figure 4 and the prompt used with the distilled model is shown in Figure 5; the prompt used for extraction of facts is shown in Figure 6; and the zero shot experiment prompt is shown in Figure 7.

**Entity extraction.**   When generating distillation data for entity tagging using Llama 3.1 405B, we dynamically selected two few-shot examples to accompany each target paragraph. This process began by embedding the target paragraph, along with a set of "golden" paragraphs (those with known tags), using `text-embedding-3-large`. The first example was chosen as the nearest neighbor to the target paragraph from within the golden paragraphs that contained one or more entities. The second example was selected as the nearest neighbor to the target paragraph verified to contain no entities. These two examples were then included in the prompt to the model, alongside the target paragraph itself, to guide the entity tagging process. After collecting a set of labeled data, we discard the large prompt with few shot examples and distilled into Llama 3.1 8B using the prompt shown in Figure 5.

**Fact extraction.**   For fact extraction, we provide the model with four static few-shot examples, two of which contain paragraphs that have relevant facts about entities, and two that do not. One example of each is shown in Figure 8.

**Zero-shot.**   For generating positive and negative examples in our zero-shot setting, the prompt (Figure 7) requires task-specific descriptions for the positive and negative classes. These descriptions are straightforward translations of the task objective. For example, when working with the `BBB Martins` dataset (which classifies drugs by their ability to cross the blood-brain barrier), the text for `<TASK POSITIVE DESCRIPTION>` is "crosses the blood brain barrier," and the `<TASK NEGATIVE DESCRIPTION>` is "does not" (understood in context as "a compound that does not cross the blood brain barrier"). Similarly, for `AMES`, the positive and negative texts are "is mutagenic" and "is not mutagenic".

## B   Dataset Statistics

Our release contains two sub-corpora: facts about small molecules and facts about genes/proteins. These were produced after (i) document retrieval, (ii) paragraph-level entity tagging, (iii) entity normalization, and (iv) fact extraction (Section 3). Full counts of unique papers, passages, entities, and passage-entity pairs (up to step (iii)) are provided in Table 5, broken down by small molecules and genes/proteins.

To make fact extraction feasible, `Medex` is generated from a subset of the full corpus. To create this subset, subsampling was performed independently for small molecules and genes/proteins. For each entity type, we iteratively looped through the unique entities; in each pass, one paragraph was randomly selected for an entity from its associated pool and added to our working set. This iterative selection continued until over 6 million paragraphs were gathered for that specific entity type. This approach was also intended to mitigate the over-representation of well-studied molecules and proteins. This process resulted in a working set of approximately 12 million paragraphs (roughly 6 million per category).

Extraction of facts from the working set yielded approximately 16 million facts across roughly 900K small molecules and around 16 million facts for 327K protein/gene entities. This resulted in a median of 2 facts per unique small molecule and 4 facts per unique protein/gene in `Medex`. In the future, we plan on providing a release drawing from a significantly larger portion of the corpus.

# C  Optimization experiment details

To produce the results shown in Figure 3, we ran Bayesian optimization (BayesOpt) with and without a toxicity classifier as a hard constraint. In particular, we ran LOLBO [53] as it is a popular SOTA BO method for computational drug design. For constrained BO, we ran LOLBO with the standard SCBO [13] method to impose a hard feasibility constraint. In each case, we ran with a budget of 200,000 black box function evaluations and used all of the same default hyperparameters as in the original LOLBO paper [53].

In this experiment, we defined toxicity as a combination of (a) whether the compound may cause hERG channel blockade and (b) the compounds potential for mutagenic effects. Classification was implemented with calibrated [21] KNN classifiers over `MedexCLIP` embeddings using the TDC training data for the `AMES` and `hERG Karim` tasks. If either classifier reported a likelihood of toxicity greater than $30\%$ for a given compound, it was considered toxic and rejected.

# D  Compute resources

In this section, we provide details about all compute resources used to produce results provided in this work.

**Compute specifications.**    We use GPUs to run all experiments and produce all results provided. Our internal GPU cluster consists of 2 GPU nodes with 10 NVIDIA RTX A5000s each, and 9 GPU nodes with 8 NVIDIA RTX A6000s each. We supplemented our nodes with A6000 workers from `runpod.io`.

**Execution time.**    For optimization results provided in Figure 3, we utilized roughly 700 total GPU hours on our internal cluster. We estimate that entity tagging and fact extraction took approximately 5,500 GPU hours. Training TDC LM, `MedexCLIP`, `MedexLLaVA`, and `MedexLM` took a collective 1008 GPU hours. Thus, completing all experiments needed to produce the results provided in this paper required roughly 7,208 total GPU hours. Preliminary experiments required additional compute.

# E  Architectures and hyperparameters

Below we provide a detailed account of the architectures and relevant hyperparameters for the models in this work. All models are optimized using Adam [35], and all MLPs use the GeLU activation [24]. We provide checkpoints for `MedexCLIP` and code to reproduce the main results in the supplemental materials.

`Molecule Encoder.`    We use a standard encoder-decoder T5 model [63] with a hidden size of 256, trained on sequences from PubChem [34] and ZINC20 [31], with a maximum sequence length of 768. SMILES were converted to Kekulé form and were tokenized with a method closely resembling the SMIRK tokenizer [76].

`Protein Encoder.`    We use the ProtT5-XL model described in [12].

`MedexCLIP.`    For contrastive pretraining, we use two 2-layer MLPs: one for the structure (small molecule or protein embedding produced by their respective encoder), and one for the text input. The MLP for small molecules has a hidden dimension of 1536, while the protein MLP has a hidden dimension of 1024. Each MLP projects into a joint 128-dimensional embedding space. A learnable temperature parameter $\tau$ is initialized to $0.1$ and is learned jointly with the MLPs.

`TDC LM.`    TDC LM is a Qwen 2.5 0.5B [61] base model finetuned on the selected tasks from TDC. For training, we use a batch size of 256, a peak learning rate of $4 \times 10^{-5}$, 1024 warm-up steps, and cosine decay to zero.

`MedexLM.`    MedexLM is identical to TDC LM, but the TDC training dataset was augmented with facts extracted from `Medex` that overlap with the tasks in TDC. Training hyperparameters are identical to those used in TDC LM.

Table 6: TDC regression benchmarks. Underline: SOTA, bold: best generalist.

| Task | Metric | Specialist SOTA | MedexCLIP | TxGemma 2B | TDC LM |
|------|--------|-----------------|-----------|------------|--------|
| BindingDB Patent | PCC | 0.588 [39] | **0.691** | 0.422 | 0.300 |
| BindingDB ic50 | Spearman | 0.637 [36] | **0.813** | 0.399 | 0.548 |
| BindingDB kd | PCC | 0.712 [32] | **0.697** | 0.352 | 0.450 |
| BindingDB ki | PCC | 0.840 [80] | **0.775** | 0.661 | 0.589 |
| Buchwald Hartwig | PCC | 0.786 [60] | **0.921** | 0.861 | 0.707 |
| Caco2 Wang | MAE | 0.285 [27] | **0.382** | 0.476 | 0.528 |
| Clearance H. AZ | Spearman | 0.440 [65] | **0.467** | 0.353 | 0.153 |
| Clearance M. AZ | Spearman | 0.625 [27] | **0.597** | 0.468 | 0.387 |
| DAVIS | MSE | 0.219 [57] | **0.541** | 0.601 | 0.790 |
| DisGeNET | MAE | N/A | 0.058 | **0.057** | 0.081 |
| DrugComb Bliss | MAE | 4.560 [83] | 3.877 | 4.230 | **3.715** |
| DrugComb CSS | MAE | 16.858 [83] | 8.296 | 15.752 | **7.748** |
| DrugComb HSA | MAE | 4.453 [83] | 3.716 | 4.231 | **3.538** |
| DrugComb Loewe | MAE | 9.184 [83] | 7.016 | 17.342 | **6.850** |
| DrugComb ZIP | MAE | 4.027 [83] | 3.172 | 3.950 | **3.065** |
| GDSC1 | PCC | 0.860 [48] | **0.886** | 0.876 | 0.872 |
| GDSC2 | PCC | 0.860 [48] | **0.879** | 0.824 | 0.865 |
| Half Life Obach | Spearman | 0.547 [14] | **0.440** | 0.386 | 0.051 |
| KIBA | MSE | 0.154 [57] | **0.567** | 0.588 | 0.840 |
| LD50 Zhu | MAE | 0.552 [27] | **0.708** | 0.710 | 0.746 |
| Lipophilicity A. | MAE | 0.467 [85] | 0.771 | **0.610** | 0.871 |
| OncoPolyPharm. | PCC | 0.730 [59] | **0.588** | 0.473 | 0.391 |
| PPBR AZ | MAE | 7.788 [85] | **7.808** | 9.266 | 9.682 |
| Protein SAbDab | MAE | N/A | 2.338 | **1.066** | 3.630 |
| Solubility AqSolDB | MAE | 0.761 [85] | 1.070 | **0.961** | 1.085 |
| TAP | MAE | N/A | **3.672** | 5.301 | 6.144 |
| USPTO Yields | PCC | 0.361 [60] | **0.548** | 0.011 | 0.272 |
| VDss Lombardo | Spearman | 0.627 [4] | 0.509 | **0.564** | 0.434 |

`MedexLLaVA.` LM architecture and initialization identical to `TDC LM`. `MedexLLaVA` replaces SMILES strings with an additional 2-layer MLP that projects `MedexCLIP` embeddings into the token embedding space of TDC LM, injecting the literature-informed small molecule representations directly into the model. Training hyperparameters are exactly the same as TDC LM. We use a hidden dimension of 2048 within the MLP.

`Supervised heads.` For downstream prediction tasks we feed the *final hidden layer* of the appropriate `MedexCLIP` MLP into a single-hidden-layer MLP (hidden dimension of 512). For tasks that have a textual input (i.e. cell line information in `DrugComb` tasks), we embed the text using `MedexCLIP` and concatenate it along with the structure embedding(s) as input to the supervised head. We use the relevant supervised heads to do the filtering shown in Figure 1.

# F   Broader impacts

**Opportunities.** `Medex` efficiently synthesizes experimentally-validated information from biomedical literature into short, self-contained facts. This enables small multimodal models (i.e. 15M trainable parameters in our case), to compete with or surpass 2B or 9B parameter models—lowering the computational barrier for academic groups working on ML aided therapeutic design. By improving the predictive performance of models (Section 5.2), and the efficiency and safety of in-silico design (Section 5.5), `Medex` has the potential to accelerate therapeutic discovery and design.

**Potential Risks.** In its current form, `Medex` inherits the biases present in biomedical literature: over representation of well studied proteins, small molecules, and associated diseases / disorders; bias towards positive results inherent in publishing; and so on. Further, extracted facts are not weighted by provenance or experimental quality—rather, each excerpt we extract data from is treated as a source

Table 7: TDC classification benchmarks. Underline: SOTA, bold: best generalist.

| Task | Metric | Specialist SOTA | MedexCLIP | TxGemma 2B | TDC LM |
|---|---|---|---|---|---|
| AMES | AUROC | 0.871 [74] | **0.802** | 0.796 | 0.759 |
| BBB Martins | AUROC | 0.915 [16] | **0.881** | 0.864 | 0.758 |
| Bioavailability Ma | AUROC | 0.748 [3] | 0.661 | **0.715** | 0.572 |
| CYP1A2 Veith | AUPRC | 0.900 [58] | **0.930** | 0.910 | 0.903 |
| CYP2C19 Veith | AUROC | 0.890 [58] | 0.888 | **0.905** | 0.868 |
| CYP2C9 S.C.M | AUPRC | 0.441 [74] | 0.430 | **0.457** | 0.426 |
| CYP2C9 Veith | AUPRC | 0.839 [26] | 0.778 | **0.801** | 0.749 |
| CYP2D6 S.C.M | AUPRC | 0.736 [26] | **0.720** | 0.605 | 0.615 |
| CYP2D6 Veith | AUPRC | 0.739 [26] | **0.648** | 0.637 | 0.594 |
| CYP3A4 S.C.M | AUROC | 0.662 [30] | **0.730** | 0.669 | 0.593 |
| CYP3A4 Veith | AUPRC | 0.904 [26] | 0.842 | **0.844** | 0.791 |
| Carcinogens L. | Accuracy | 0.770 [38] | **0.857** | 0.821 | 0.822 |
| ClinTox | AUROC | 0.948 [45] | 0.789 | **0.810** | 0.677 |
| DILI | AUROC | 0.925 [74] | **0.934** | 0.875 | 0.718 |
| HIA Hou | AUROC | 0.988 [27] | **0.986** | 0.937 | 0.901 |
| HIV | AUROC | 0.851 [44] | **0.806** | 0.737 | 0.744 |
| HuRI | AUPRC | 0.724 [64] | 0.712 | **0.751** | 0.622 |
| MHC1 IEDB | AUROC | 0.986 [19] | 0.861 | **0.910** | 0.887 |
| MHC2 IEDB | AUROC | 0.940 [54] | **0.852** | 0.812 | 0.849 |
| PAMPA NCATS | AUROC | 0.900 [68] | **0.769** | 0.642 | 0.654 |
| Pgp Broccatelli | AUROC | 0.935 [74] | 0.896 | **0.900** | 0.896 |
| SARSCoV2 3CLPro | AUROC | 0.800 [22] | 0.711 | **0.733** | 0.561 |
| SARSCoV2 Vitro | AUROC | 0.640 [51] | 0.556 | **0.650** | 0.367 |
| SAbDab Chen | AUPRC | 0.510 [8] | 0.659 | **0.676** | 0.616 |
| Skin Reaction | AUROC | 0.840 [2] | 0.649 | **0.671** | 0.529 |
| Tox21 | AUROC | 0.961 [67] | 0.880 | **0.881** | 0.870 |
| ToxCast | AUROC | 0.777 [45] | **0.880** | 0.784 | 0.880 |
| Butkiewicz | AUROC | 0.840 [75] | **0.874** | 0.791 | 0.809 |
| hERG | AUROC | 0.874 [3] | **0.885** | 0.876 | 0.878 |
| hERG Karim | Accuracy | 0.770 [33] | 0.757 | **0.778** | 0.720 |
| herg central | AUROC | 0.860 [37] | 0.877 | **0.880** | 0.870 |
| phase1 | AUROC | 0.576 [17] | 0.579 | **0.642** | 0.612 |
| phase2 | AUROC | 0.645 [17] | 0.618 | **0.665** | 0.659 |
| phase3 | AUROC | 0.723 [17] | 0.696 | **0.731** | 0.712 |
| Weber | AUROC | 0.870 [79] | 0.582 | **0.730** | 0.538 |

of truth—meaning downstream models may inherit any spurious findings or incorrect assertions within the initial corpus. Finally, `Medex` has the potential to be coupled with generative models to assist in the development of dual use or illicit substances.

# G   Full Results

Tables 6 and 7 report per-task numbers for all 63 Therapeutic Data Commons (TDC) benchmarks. For completeness, they include an LLM trained exclusively on TDC, confirming that the improvements are not merely architectural but stem from pretraining on `Medex`. We highlight a few takeaways:

**Parameter-efficient performance.**   `MedexCLIP` (15M trainable parameters on top of frozen encoders) matches or exceeds the much larger TxGemma-2B on *23/28 regression* tasks, reducing the average MAE by 33%, and raising mean performance on classification from 0.768 to 0.771.

**Broad coverage.**   Performance gains are not confined to toxicity; they extend to pharmacokinetics, reaction yields, protein interaction, drug synergy, and more. This breadth confirms that literature-distilled priors encode a wide array of high quality information, benefiting the diverse tasks within TDC.

**Zero-shot capabilities.** Without any TDC fine-tuning, the same `MedexCLIP` encoder attains a mean AUROC of $0.718$ across nine safety and ADME related assays, a $74\%$ relative lift over a 2B-parameter Gemma baseline (Table 4)—evidence that the learned joint space already organizes molecules along meaningful, text-derived axes.

```
Task: Analyze the given paragraph to identify:
1. Specific, uniquely structured small molecules and their used alternative identifiers
2. Specific, uniquely structured biologics and their used alternative identifiers
3. Classes of small molecules or biologics when statements that hold true for the entire class are
made

Definitions and Examples:
1. Small molecules: Low molecular weight compounds with defined chemical structures (generally <= 900
 daltons), things you would find on PubChem
  - Examples:
  - Individual molecules: aspirin, Leukotriene A4, 1,1,1-trifluoromethyl-6,9,12,15-eicosatetraen-2-
  one, prednisolone
  - Class extraction (when statement applies to whole class): NSAIDs, benzodiazepines, statins

2. Biologics: Large, complex molecules with defined sequences or structures (generally > 900 daltons),
 things you would find on UniProt
  - Examples:
  - Individual macromolecules: insulin
  - Proteins / peptides / large enzymes, antibodies: IL-6, TNF-alpha, rabbit anti-5-HT antibody,
  MAPK, Drosomycin
  - Class extraction (when statement applies to whole class): gonadotrophins, Karyopherins

Instructions:
1. Before tagging any entity, verify:
  - For specific molecules or biologics:
   - Is this a specific, named molecule/biologic rather than a class?
   - Would this molecule/biologic have a defined chemical structure or sequence?
   - If multiple similar molecules/biologics are mentioned, can each one be distinguished?

  - For classes:
   - Does the statement apply to the entire class?
   - Is meaningful information provided about the class as a whole?
   - Is the class specific enough to have shared mechanisms or properties?
   - Is the class not too broad or general to be useful?

  - Only proceed with tagging if the relevant answers are "yes".

2. For entities that pass the verification:
  - Identify the entity as a small molecule or biologic
  - List each entity only once, including all alternative identifiers
  - Extract entities in their singular form
  - For genes encoding proteins, annotate the protein rather than the gene
  - If several entities are mentioned together (i.e. "ERK1/2"), tag them as separate entities (ERK1,
  ERK2)
  - When listing the name and alternative identifiers, use the least ambiguous one as the name, and
  list all others as alternatives in order of increasing ambiguity

3. For each paragraph, assign one or more of the following category tags if relevant information is
present:
    - Structure/Properties: Information about molecular structure or physical/chemical properties
    - Chemistry: Details about chemical reactions or interactions
    - Pharmacology: Information on drug action, effects, or mechanisms
    - Synthesis/Formulation: Methods of production or preparation
    - Safety/Regulation: Information on toxicity, side effects, or regulatory status
    If none of these categories apply, tag as "None".

4. Output format:
  - {"categories":["cat1"],"molecules":[{"name":"name","alternatives":["alt1", "alt2"],"is_class":
  false}],"biologics":[{"name":"name","alternatives":[],"is_class":true}]}

Review the provided examples to understand the expected output format:
<fewshot examples>
```

Figure 4: Prompt used to extract entities from text using Llama 405B. Appendix A provides an overview of the few-shot prompting strategy.

```
Analyze the given paragraph to identify and categorize small molecules and macromolecules/biologics (
or classes thereof), including their synonyms.

Output format:
{"categories":["cat1"],"molecules":[{"name":"name","alternatives":["alt1","alt2"],"is_class":false
}],"biologics":[{"name":"name","alternatives":[],"is_class":true}]}
```

Figure 5: Prompt used with distilled models for entity tagging.

```
You are an expert biomedical information-extraction assistant.

----------------------- TASK -----------------------
Given:
1. paragraph - a single paragraph from a PubMed article
2. target_entities - a JSON list of molecules, proteins or genes I care about.

Return one-line JSON with a single key "facts", whose value is
a list of fact objects:

  {
    "facts": [
      {
        "entity": "<entity>",
        "fact": "<self-contained fact>",
      },
      ...
    ]
  }

--------------- WHAT COUNTS AS A FACT ---------------
A fact is a generally true, reusable property of the entity that
remains meaningful outside the paragraph.

Allowed (non-exhaustive):
- Identity or classification
- Mechanism of action / target binding
- Therapeutic or functional use
- Physiological / pathophysiological role
- Broad pharmacological / chemical property

NOT allowed (discard):
- Experimental specifics without context (e.g., "EC50 = 2.6 nM"
  with no assay, cell type, etc.).
- Study design, cohort sizes, p-values.
- Pure rank orders ("better than X") with no property named.
- Speculative language ("may", "might").
- Facts about entities not in target_entities.

------------------ REQUIRED CONTENT ------------------
If the paragraph provides quantitative values (percent uptake,
concentrations, EC50, etc.) that define the property, you must
embed those numbers (with units and key conditions, e.g., time-point
or tissue) directly in the "fact" string. Supply just enough experimental
context (assay, cell type, time) so the statement is interpretable on its own.

-------------------- OUTPUT RULES --------------------
1. JSON output, no extra keys, no Markdown.
2. Preserve exact spelling of each entity from target_entities.
3. Remove duplicate facts (case-insensitive exact match).
4. If no valid facts, output {"facts":[]}.
5. If a target entity is provided, but cannot be found in the the paragraph, ignore that entity.
6. If there are no facts for a target entity, ignore that entity.

-------------------- EXAMPLES -----------------------
<fewshot examples>
------------------- PROMPT END ----------------------
Begin.
```

Figure 6: Prompt used to extract facts from text using GPT 4.1. See Appendix A for a description of the few-shot examples.

```
Generate 10 diverse, short, 1-2 sentence facts each describing a unique compound that <TASK POSITIVE
DESCRIPTION>, and 10 facts each describing a unique compound that <TASK NEGATIVE DESCRIPTION>.
Present them as a JSON dictionary, with keys "text_pos" and "text_neg", both with lists of strings.
The diversity should lie in the different reasons for falling in text_pos and text_neg, i.e. we want
to represent diverse failure and success cases.
```

Figure 7: Prompt used to generate positive and negative facts used in zero-shot experiments. See Appendix A for a discussion of the task-specific descriptions.

```
Example 1:
{
  "paragraph": "We selected a pair of closely related analogues of which one compound, CBK006377 (
  referred to as CBK77; N-[6-ethoxy-1,3-benzothiazol-2-yl]-5-nitrofuran-2-carboxamide), displayed
  profound UPS impairment and cellular toxicity, while the second compound, CBK085907 (referred to as
   CBK07; N-(4-methoxy-1,3-benzothiazol-2-yl)-5-nitrofuran-2-carboxamide), lacked these activities.
  The EC50 of CBK77 was determined as 4.3 microM (6 h treatment, 95% C.I 3.8-5.0 microM) with no
  detectable inhibition for CBK07 in the tested concentration range. It should, however, be mentioned
   that CBK07 is not completely inert as we observed modest UPS impairment and toxicity at high
  concentrations (>50 microM) over longer incubations (24 h). The uptake of CBK77 and CBK07 in cells
  was comparable, excluding that the strongly reduced activity of CBK07 could be attributed to a loss
   of cell permeability. Together these data show that CBK77 blocks degradation of a reporter
  substrate of the UPS and induces cell death.",
  "target_entities": [
    "CBK006377",
    "N-(4-methoxy-1,3-benzothiazol-2-yl)-5-nitrofuran-2-carboxamide"
  ]
}
{
  "facts": [
    {
      "entity": "CBK006377",
      "fact": "CBK006377 impairs the ubiquitin proteasome system and induces cell death, showing an
      EC50 of 4.3 uM (6 h treatment, 95 % CI 3.8 5.0 uM) for blocking degradation of a UPS reporter
      substrate.",
    },
    {
      "entity": "N-(4-methoxy-1,3-benzothiazol-2-yl)-5-nitrofuran-2-carboxamide",
      "fact": "N-(4-methoxy-1,3-benzothiazol-2-yl)-5-nitrofuran-2-carboxamide shows no detectable
      ubiquitin proteasome system inhibition within the tested concentration range up to 50 uM (6 h),
      but produces modest UPS impairment and cellular toxicity at concentrations >50 uM after 24 h.",
    },
    {
      "entity": "N-(4-methoxy-1,3-benzothiazol-2-yl)-5-nitrofuran-2-carboxamide",
      "fact": "Despite its low UPS-inhibitory activity, the cellular uptake of N-(4-methoxy-1,3-
      benzothiazol-2-yl)-5-nitrofuran-2-carboxamide is comparable to that of CBK006377, indicating
      that reduced efficacy is not caused by poor cell permeability.",
  }
  ]
}

Example 2:
{
  "paragraph": "OBJECTIVE: To study the chemical constituents of Ligularia macrophylla. METHODS:
  Isolation and purification were carried out on repeated silica gel column chromatography. The
  structures of the compounds were identified by physico-chemical properties and spectral analyses.
  RESULTS: Eight compounds were isolated and identified as kaempferol (1), 2,4'-dihydroxy-5-
  methoxychalcone (2), 5-hydroxy-3,4', 7-trimethoxyflavone (3), isobutyl ester terephthalic acid (4),
   4-hydroxybenzaldehyde (5), mono (2-ethylhexyl) terephthalate (6), lupeol (7), beta-sitosterol (8).
   CONCLUSION: Compounds 1 - 7 are isolated from this plant for the first time.",
  "target_entities": [
    "kaempferol",
    "2,4'-dihydroxy-5-methoxychalcone",
    "5-hydroxy-3,4',7-trimethoxyflavone",
    "isobutyl ester terephthalic acid",
    "4-hydroxybenzaldehyde",
    "mono (2-ethylhexyl) terephthalate",
    "lupeol",
    "beta-sitosterol"
  ]
}
{
  "facts": []
}
```

Figure 8: Two of the static few-shot examples provided while generating distillation data for fact generation (see Figure 6 for the full prompt).

