# OpenReview forum: "A Dataset for Distilling Knowledge Priors from Literature for Therapeutic Design"
_NeurIPS.cc/2025/Datasets_and_Benchmarks_Track — NeurIPS 2025 Datasets and Benchmarks Track poster_

### Official Review · Reviewer_ihhj · 2025-06-25

**Rating:** 4
**Confidence:** 4

**Summary:**

This paper introduces the Medex dataset, which comprises biological and chemical entities—such as small molecules (represented in SMILES) and genes (identified by NCBI gene IDs)—along with their associated facts extracted from the literature. Medex enables the development of enhanced representations for molecules and genes by aligning with existing knowledge. These improved representations have been shown useful in various downstream tasks, including toxicity prediction and molecular design, demonstrating superior performance compared to models trained on the existing sota dataset (TDC).

**Dataset Code Accessibility:**

Yes

**Dataset Code Comments:**

The dataset is publicly available on huggingface as well as the inference code to reproduce the results.

**Ethical Considerations:**

No, there are no or only very minor ethics concerns

**Final Justification:**

Thank you for your responses. As most of my concerns have been addressed, I will retain a positive score.

**Limitations Weaknesses:**

1. The dataset starts by choosing entities and then pulling up related literature for each one. However, the exact way these entities were picked isn’t explained. Could the authors explain their selection process and whether this might lead to any selection bias?

2. The dataset includes both genes and proteins, but it’s not clear how the literature for genes is separated from that for the same proteins. Since gene–gene interactions can be very different from protein–protein interactions, could the authors clarify how they tell the difference between facts about a gene and those about the corresponding protein?

3. Could the authors provide evidence or reasoning to show that these distilled facts are actually more useful or valuable than the established descriptions, such as NCBI gene descriptions used in GenePT?

4. Also, adding error bars to the experimental results would make the findings more reliable. Error bars can show the statistical significance of the differences and make the performance evaluations clearer.

**Strengths Contributions:**

This paper presents a comprehensive collection of biological entities—including small molecules, genes, and proteins—with the potential to significantly impact a variety of applications. The dataset is meticulously curated to encompass a wide range of data, ensuring that it is both robust and versatile for use in diverse research domains.

The authors further substantiate the utility of the dataset by developing and evaluating several models, such as MedexCLIP. They compare these approaches with established baselines across multiple downstream tasks, demonstrating consistent and significant improvements. This comparative analysis highlights the enhanced performance and broader applicability of the dataset in practical scenarios.

---

> ### Author Rebuttal · Authors · 2025-07-30
>
> Thank you for your review. We will try to clarify here how we went about choosing entities and papers, and we will update the paper to be more clear. PubChem and UniProt curate molecules and proteins respectively, and they both associate their entities with papers when available. For example, PubChem has a file listing every entity-paper pair they are aware of (you can find it on their FTP server). We collected all available papers listed by PubChem and UniProt, then ran our entity tagger over each so we could get entity tags at a paragraph granularity. This then became our set of (entity, paragraph) pairs that we used to distill facts from. We took this approach as it gave us some level of confidence that the given paper discussed at least one molecule or gene / protein, though most papers from this set discussed several.
>
> In its current form, Medex provides no delineation between genes and proteins. The normalizer we used (GNorm2) simply provides an NCBI Gene ID for a given annotation. This is a clear limitation with how we are handling genes / proteins, though we find that our models perform well despite it. We likely could infer whether the paragraph / fact is discussing the gene or protein based on context, and it may be a worthwhile effort to do a pass over the dataset to build that separation.
>
> We believe that the facts within Medex are complementary to the summaries that one can find in resources such as NCBI Gene, ChEBI, Uniprot, or PubChem. For example, the summary provided by NCBI Gene for TP53 (NCBI 7157) is high quality, but it is simply that: a summary. Medex collates knowledge directly from the literature, at an incredibly large scale, allowing for more thorough and nuanced information regarding these entities. For example, some of the facts that Medex includes about TP53 are:
> - TP53 represses glycolysis by inhibiting the transcription of PFKFB3 and PFKFB4 genes, reducing fructose-2,6-bisphosphate expression. [1]
> - In malignant fibrous histiocytoma (MFH), p53 immunohistochemical positivity is associated with a missense mutation of the TP53 gene and serves as a marker of neoplastic cells. [2]
> - TP53 mutations occur in 15% of ovarian clear cell carcinomas. [3]
>
> Likewise for small molecules, many entities simply do not have information about them available in repositories like ChEBI or PubChem, existing solely within the literature. Even extremely common molecules will be missing most (by quantity) of our knowledge about them. A summary of aspirin will obviously mention its use in pain relief and maybe even that it is a mild blood thinner but might not, for example, mention its specific effective use to prevent blood clots in infants after a BT shunt surgery.
>
> To investigate this further, we used an LLM to analyze 20,000 facts from Medex (1,000 for each of 20 selected genes). The goal was to determine if a fact's information was already contained in its corresponding NCBI Gene summary, or if it presented new information. We found that only 233 (~1%) of these facts presented no new information outside of the summary, suggesting that the majority of facts within Medex provide unique knowledge not found in curated summaries about the entities.
>
> [1] Czegle, I., Gray, A. L., Wang, M., Liu, Y., Wang, J., & Wappler-Guzzetta, E. A. (2021). Mitochondria and Their Relationship with Common Genetic Abnormalities in Hematologic Malignancies. Life, 11(12), 1351.
>
> [2] Szollosi, Z., Nemeth, T., Egervari, K., & Nemes, Z. (2005). Histiocyte-like cells expressing factor XIIIa do not belong to the neoplastic cell population in malignant fibrous histiocytoma. Pathology, research and practice, 201(5), 369–377.
>
> [3] Kuo, K. T., Mao, T. L., Jones, S., Veras, E., Ayhan, A., Wang, T. L., Glas, R., Slamon, D., Velculescu, V. E., Kuman, R. J., & Shih, I.eM. (2009). Frequent activating mutations of PIK3CA in ovarian clear cell carcinoma. The American journal of pathology, 174(5), 1597–1601.

---

> > ### Comment · Reviewer_ihhj · 2025-08-05
> > **Rebuttal response**
> >
> > Thanks for the authors'  responses and most of my concerns have been addressed.

---

### Official Review · Reviewer_NwMR · 2025-06-27

**Rating:** 5
**Confidence:** 4

**Summary:**

AI based simulations of molecules often produce results that we find undesirable in practice. The authors propose a new dataset of curated knowledge about health-related-chemistry and biochemistry. The goal is to train models specifically on this dataset so they are less likely to break safety or desirability constraints, and are more likely to produce results that are useful to medical researchers. They focus on incorporating text-based information rather than using only structural and scientific records, since much of the literature and existing information regarding drug safety is found primarily in text rather than scientific databases.

The value of their contributions is tested via the widely used GuacaMol testing benchmark.

**Additional Feedback:**

I am happy to give this paper an accept due to the strong contributions of the content. While I do not view them as a final product, the sample networks you train make a strong argument for the usefulness of the dataset. Additionally, reading about and considering the description of the dataset that you provide leaves me feeling confident that this will be a potentially very valuable dataset to the community.

I strongly encourage the authors to check the downloading and metadata associated with the Hugging Face upload. I expect that this will be fixed and made to match the neurips required standard for this year’s submissions.

Lastly, I encourage the authors to make some revisions on the document to improve readability and clarity. I can see that since you are already familiar with the content you tend to gloss over some details or jump back and forth between ideas. Limitation (5) addresses this. I would recommend perhaps asking somebody unfamiliar with the project to read it and map out how they think every paragraph connects. This can provide valuable information regarding how to rearrange content so that everything related to one topic is grouped together.

**Dataset Code Accessibility:**

Partly

**Dataset Code Comments:**

(1) The provided Hugging Face download code is:
 from datasets import load_dataset
dataset = load_dataset("datasets/medexanon/medex")
dataset = dataset["train"]  # or "test", "validation" as needed
print(dataset)

However, the “dataset = load_dataset(“datasets/medexanon/medex”)” line should not have the “datasets/“ in it. This must be removed in order for it to be downloaded correctly.

Additionally, the code:

import mlcroissant as mlc
croissant_dataset = mlc.Dataset("https://huggingface.co/datasets/medexanon/medex/resolve/main/croissant.json")
print(croissant_dataset.metadata.record_sets)

Does not work properly and produces a JSONDecodeError. This needs to be addressed before the final version is released.

(2) I could not find the core metadata presence. Neither the automated checker nor my manual verification could locate this. It should include basic info like name and licensing information.

**Ethical Considerations:**

No, there are no or only very minor ethics concerns

**Final Justification:**

The authors contribute a substantial new amount of data and provide data curation to ease the integration of other existing similar datasets. They show small scale experiments which empirically supports the usefulness of the data. My main issue with the submission was the download link and missing metadata: both issues have been resolved in time for the rebuttal.

Additionally, the authors took some suggestions on improving clarity and readability of the paper, making it easier for users to find what they need in the writeup.

Overall, this work matches exactly what we look for in a datasets work... they introduce a substantial new dataset and show that the use of their dataset can immediately lead to improved results on existing benchmarks. I do not believe it to be a groundbreaking work or a revolutionary paper, but it is a strong and useful contribution to literature.

**Limitations Weaknesses:**

(1) It should be made clear in the Title and Abstract the domain of the knowledge base. From the Title and the beginning of the Abstract, it sounds like this is a dataset for all scientific fields. This leads to confusion when biomolecules, initially seeming to be an example, turn into the focus of the paper.

(2) While effective, the near exclusive use of already existing models to select and curate the dataset reduces the novelty of the approach. Small adapters to match the problem domain do little to assuage this.

(3) Grammer: in line 194 it should be “works” not “work”.

(4) In line 204, do you not allow for multi-class classification? Every classification problem is binary?

(5) Section 4 is somewhat hard to follow. The inclusion of so many different variations of the model one after another is not ideal. Maybe a table showing what each has, or even just a list with a short summary could help with clarity and readability. Furthermore, combining Sections 4 and 5 or at least making a clear link between the related parts of the two would be beneficial.

**Strengths Contributions:**

(1) Great size of the dataset… 200 million passages tied to 2 million entities is a substantial amount for machine learning. A total of 36.4 LLM curated facts is also significant.

(2) Justification of value from preliminary results is strong. Both production of less error in predicted results as well as fewer toxic results demonstrates the validity of this work.

(3) The entity-first filtering process for selecting text based resources helps to reduce irrelevant information. It can be adapted to analogous ideas in other fields as well.

---

> ### Author Rebuttal · Authors · 2025-07-30
>
> Thank you for your thorough review. You’re right in that our title and abstract allude to a general purpose dataset, while we primarily focus on therapeutic design. The PCs recently disallowed PDF edits, but for the camera ready version we promise to update the title and abstract to more accurately reflect the scope of Medex, e.g. by modifying “for Scientific Discovery” to “for Molecular Design” or similar. We think something along these lines is reasonable – while the writing of our paper focuses on therapeutic design as a motivating example, the fact set was extracted broadly and will contain facts from chemistry more broadly than solely medicinal chemistry specifically (and at least some of the TDC facts focus on more broad topics like yield prediction). Let us know what you think, however – we’re open to “Therapeutic Design” exclusively given the focus of most of the experimental validation.
>
> With respect to loading the dataset and croissant information, we agree with you and will improve the clarity of our instructions. **We have updated the Medex dataset page on HuggingFace with instructions for both, and we will also be publishing a GitHub repository that will have instructions and example code as well.** For croissant specifically, the file is already live right now on the HuggingFace dataset page but the URL used seems to be in error, which we cannot link due to author discussion phase rules this year. The correct croissant file was also included with our submission, which we (as authors at least) can download directly from OpenReview but perhaps they didn’t make it available or prominent to reviewers. We’ve verified that your code works with the correct URL to the croissant file (which we again apologize that we can’t just include here). **Likewise, we will update the dataset / croissant with missing information (such as license).**
>
> Finally, we agree that Section 4 could be made more clear. For the camera-ready version we will update Section 4 with a table containing clear model descriptions, and ensure the rest of the section and the broader paper is clear and easy to follow.

---

> > ### Comment · Reviewer_NwMR · 2025-08-01
> > **Response to the Rebuttal**
> >
> > I appreciate the authors' willingness to change the title. I am completely fine with "Molecular Design" if that is the authors preference, especially if they include the same explanation that is provided in the rebuttal.
> >
> > I appreciate the updates to the dataset pages. The inclusion of a GitHub repo with example usage is very helpful as well. I understand the difficulties with the url transfer... it is an unfortunate reality of the review situation.
> >
> > Lastly, I appreciate the willingness to make clarity updates to Section 4.

---

### Official Review · Reviewer_m9S7 · 2025-06-28

**Rating:** 5
**Confidence:** 4

**Summary:**

The paper proposed a new dataset, Medex, a dataset of priors extracted from literature describing compounds used in lab settings. This dataset can bridge the molecular structure and the literature text with the help of 3.7M scientific papers. They perform fine-grained paragraph-level entity tagging by extracting the entity in a paragraph and replacing the entity with a string like SMAILES, finally building facts for this entity using LLMs. After that, they demonstrate the effectiveness of the proposed dataset and SOTA result on multiple tasks in TDC benchmark.

**Additional Feedback:**

After reading the rebuttal, I tend to increase my score.

**Dataset Code Accessibility:**

No

**Dataset Code Comments:**

The dataset should also attach the used evaluation code using LLMs and CLIPs.

**Ethical Considerations:**

No, there are no or only very minor ethics concerns

**Final Justification:**

The response is convincing to address the dataset quality concern.

**Limitations Weaknesses:**

1. My main concern is the creation of facts for entity-fact pairs using distillation.  Although the paper used a validation set and demonstrated a good precision and recall rate of 0.922 and 0.919, respectively, the paper did not provide a detailed analysis of the wrong distillation facts, which can provide insight and bias for training large models. For example, what is the effect given that the dataset has nearly 8% bad pairs?

2. Unclear description, such as in lines 231 and 240, I did not get the example of the used instruction tuning dataset and idea of creating a multiple choice question.

3. Some presentation issues, like that Figure 3 should be a Table 3.

**Strengths Contributions:**

Overall, the proposed dataset has some novelty that builds a large-scale text prior for a bunch of molecules, making it potentially useful for AI for science, especially for training large LLMs.

Extensive validation on multiple models like CLIP, getting new SOTA results on a popular benchmark with 35 binary classification tasks and 28 regression tasks.

---

> ### Author Rebuttal · Authors · 2025-07-30
>
> Thank you for your review. Concerning the precision and recall of entity tagging, although the raw evaluation shows a precision of 0.922, the majority of the flagged false positives never enter the dataset. We manually inspected 50 entities that our distilled model tagged but Llama 405B did not. Of these, 41 (82%) referred to entities such as cell types or genes–things that cannot be normalized to SMILES strings. The remaining 9 instances (18%) were true mistagged molecules that made it past normalization. Extrapolating this, only 18% of the 8% gap corresponds to real errors, giving an effective error rate of roughly 1.4% with an adjusted precision of ~0.986. We believe the strong downstream performance of models trained on Medex support this estimate, and we will clarify the filtering effect in the paper.
>
> Nonetheless, the quality of the data within Medex is a primary concern of ours and we are actively investigating methods to both improve and verify the dataset. This includes looking at how many papers corroborate a given fact, impact factor of the journal the paper was published in, and so on. Further, we are working on pipelines using more advanced reasoning models like GPT o3 to verify the individual facts to ensure quality across the dataset. We expect that as the dataset is improved in this regard, the downstream performance of models trained on Medex will improve.
>
> Finally, we agree that the clarity in Section 4 could be improved. Before camera-ready we will ensure that the clarity of Section 4, and any other areas that might need work, will be updated accordingly. Likewise, thank you for pointing out the mistake with Figure 3, we will correct that.

---

### Official Review · Reviewer_eWiZ · 2025-07-02

**Rating:** 5
**Confidence:** 3

**Summary:**

By extracting text and facts from scientific literature with LLM, this work releases Medex, a dataset of medical entities, associated text and 36.4 M extracted facts. Leveraging this data, authors train small multimodal models with 15M learnable parameters, which outperforms Larger models on various TDC tasks, especially in zero-shot settings. Furthermore, this model can be used to generate constrains for bayesian optimization method for molecule generation, leading to safer molecules with nearly the same score.

**Dataset Code Accessibility:**

Partly

**Dataset Code Comments:**

The link only contains dataset, while models are missing.

**Ethical Considerations:**

No, there are no or only very minor ethics concerns

**Final Justification:**

The new dataset contains rich information from scientic literatures, which can benefits downstream AI4Science tasks significantly. Moreover, rebuttal also resolve my concerns. Therefore, I keep my score.

**Limitations Weaknesses:**

1. Some scientific literature are not reliable, so filtering based on number of citations, and reputation of author is needed.

2. Though smile can represent molecules, graph method often outperforms, due to the inductive bias of graph structure. Therefore, adding graph as one extra modality may further improve model's performance.

**Strengths Contributions:**

1. Large and comprehensive dataset. With  214M tagged paragraphs, and 619M aligned(entity,paragraph) pairs.
2. Carefully-designed entity normalization and alias resolution strategy, making the dataset easier to learn.
3. Effective strategy for distilling small models to save LLM expense.
4. Strong and lightweight multimodal model.

---

> ### Author Rebuttal · Authors · 2025-07-30
>
> Thank you for your thoughtful review. We agree that not all papers are as rigorous as one would want from a publication; this is a primary concern for us, and we’ve been exploring methods of further validation of the facts. **In the short term, we have updated the dataset to include information about the journal (name, ISSN, and eISSN where available), and are working on retrieving citation counts for all papers.** Citation count requires going through an API (such as OpenCitations), which will take time, but we expect to have this done by the end of August. Our medium term efforts are focusing on strategies like using a more expensive but longer reasoning LLM like GPT o3 to fact check a large fraction of the dataset, allowing us to produce estimated error statistics. More broadly, we are focused on fact corroboration in other papers, journal impact factor, author reputation, and so on, as future work to further improve the quality of facts in Medex. As we develop these methods, we will update the dataset with verification statuses for each individual fact.
>
> Graph neural networks indeed often outperform “string” based methods for small molecule property prediction and exploring the architecture best capable of exploiting / representing the information within Medex is an important next step for us. In this paper specifically, our main goal with experimental validation was to demonstrate that the dataset itself has obvious value, and therefore chose simple, easy to train models to demonstrate the effectiveness of the data.
>
> With respect to dataset code availability, we attached the inference code / models and steps to run in the supplemental information. The visibility of this is less than desirable, so we will release code and instructions in a publicly accessible GitHub repository that we will continue to maintain once broader anonymity is less of a concern.

---

### Decision · Program_Chairs · 2025-09-18

**Decision:**

Accept (poster)

**Comment:**

This submission introduces Medex, a large-scale dataset of biomedical and chemical entities with 36M extracted literature-based facts, aligned to structured identifiers such as SMILES and gene IDs. The dataset is used to train lightweight multimodal models that outperform much larger baselines on Therapeutic Data Commons tasks and improve molecule generation safety on GuacaMol.

All reviewers recommend acceptance, recognizing the dataset’s significant utility and impact for AI-driven scientific discovery. While presentation can be improved and further validation of data quality would strengthen the work, the contribution aligns well with the goals of the Datasets & Benchmarks track at NeurIPS. AC thus recommends acceptance. It is a very useful contribution to the field of AI for science.

===== FINAL UPDATE FROM DB Track PCs ====

The final decision for this paper has been taken by the program chairs after consultation with the SACs. All Senior Area Chairs have ranked papers according to the feedback from the AC during the review process. We decided to leave the original meta-review to reflect the opinion of the AC in light of the initial discussions with reviewers and SAC.